# Cognitive Foraging Enrichment (but Not Non-Cognitive Enrichment) Improved Several Longer-Term Welfare Indicators in Bottlenose Dolphins

**DOI:** 10.3390/ani13020238

**Published:** 2023-01-09

**Authors:** Isabella L. K. Clegg, Mariana Domingues, Elin Ström, Linda Berggren

**Affiliations:** 1Animal Welfare Expertise, Winchester SO22 6QU, UK; 2Groningen Institute for Evolutionary Life Sciences (GELIFES), University of Groningen, 9700 CC Groningen, The Netherlands; 3Kolmårdens Djurpark, 618 92 Kolmården, Sweden

**Keywords:** animal behaviour, bottlenose dolphins, cognitive enrichment, environmental enrichment, zoo animal welfare

## Abstract

**Simple Summary:**

Cognitive foraging enrichment is not commonly provided to dolphins in zoos, and research with other species suggests the potential for significant welfare benefits. We provided 11 bottlenose dolphins (*Tursiops truncatus*) at Kolmårdens Djurpark with alternating weeks of only cognitive enrichment, or only non-cognitive enrichment devices over an 8-week study period while recording data from several multidimensional welfare-related parameters. We found that the dolphins were more engaged by cognitive enrichment items, both when measured through qualitative caretaker scores and quantitative behavioural observations. We also found longer term welfare-related changes: during cognitive enrichment weeks, dolphins were more motivated for training sessions, and showed less anticipatory and stereotypic behaviour. We recommend that goal-oriented cognitive foraging enrichment is prioritised by dolphin facilities, and discuss how this might best be achieved in practice.

**Abstract:**

Bottlenose dolphins are the most common cetacean kept globally in zoos and aquaria (hereafter zoos), and are gregarious animals with a mostly opportunistic, generalist feeding strategy in the wild. In zoos, they have limited to no opportunities to express natural foraging behaviours as they receive their daily food ration of dead fish in a series of training sessions. Enrichment provision has increased in recent years, but items are still predominantly simple and floating in nature, and do not always target the animals’ problem-solving or food-acquisition behaviours. These discrepancies run concurrently with the intense debate about dolphin welfare in zoos and how to improve it. The current study used a within-subject design on 11 bottlenose dolphins at Kolmårdens Djurpark and measured how several welfare indicators differed between two treatments of “cognitive” and “non-cognitive” food-based enrichment. The treatments were provided on an alternating basis for eight consecutive weeks: during cognitive enrichment weeks, the animals received items which stimulated their problem-solving and foraging behaviours, and during non-cognitive enrichment weeks, they received simple items paired with fish (to eliminate bias due to food value). Data were taken related to several multidisciplinary welfare parameters during enrichment provision and training sessions, and to activity budget behaviours throughout the week. During the cognitive as opposed to non-cognitive enrichment weeks, the dolphins engaged more with the enrichment, were more motivated to participate in training sessions and performed less anticipatory and stereotypic behaviours, suggesting that cognitive enrichment improved several indicators of bottlenose dolphin welfare. Valuable lines of further investigation would be to understand how individual differences and different types of cognitive enrichment impact potential welfare benefits. Our results suggest that enrichment items promoting cognitive foraging behaviours may improve dolphin welfare, and therefore zoos might prioritise giving cognitive enrichment to this species as well as considering the same for other species with similar cognitive skills and foraging ecologies.

## 1. Introduction

A central and relatively recent goal of modern zoological parks and aquaria (hereafter zoos) is to give the animals the opportunity to express a wide range of their natural behaviour repertoire, as this has been shown to lead to improved welfare as well as visitor learning and satisfaction [1,2,3]. The efforts to achieve this goal vary between zoos, which are diverse in terms of how naturalistic their enclosures are, the extensive versus intensive management styles, and the resources spent on enrichment provision. Enclosures that are more similar to the natural environment, have less anthropocentric management and an advanced enrichment program are thought to increase choice and control over the environment and therefore animal welfare [3,4,5].

Foraging for food items or hunting prey is one of the most fundamental natural behaviours for any wild animal, and in zoos, this behaviour is always limited in some way, but to different extents depending on the species and environment [6,7]. Many studies across a range of species confirm that welfare is reduced when foraging behaviour is severely limited compared to the wild behavioural repertoire, with most demonstrating this through the improvement of welfare when opportunities to forage more naturally are given via environmental enrichment activities [6,8,9,10,11,12,13]. One hypothesis suggests that “foraging loops”, and/or those for whom such behaviour sequences constitute a large part of their wild activity budget, are more likely to experience poor welfare in zoos if they cannot forage in a similar way [9,14].

Environmental enrichment is broadly defined as any item or activity that increases the quality or value of the environment to the animal [15], and by association, enhances animal welfare [16,17]. Up until the later 20th Century, when enclosures were more barren and their hygiene was prioritised, enrichment efforts were minimal and added items were less likely to promote natural behaviours [18,19,20]. As our understanding of the animals and their needs increased, so did our efforts in enrichment, and some zoos focussed on enhancing the enclosure features while others paired enrichment with operant conditioning to promote natural behaviours [21]. Moving into the 21st Century, zoo enrichment programs have become better structured and funded, where the key principle is to develop “goal-oriented” items or activities that target a specific behaviour found to be lacking from the animals’ repertoire, and where the ultimate goal is to enhance welfare [19,22,23,24]. While positive reinforcement training, and especially cognitive training, is sometimes included with enrichment (and can certainly improve zoo animal welfare [21,25,26]), training is not considered an enrichment activity in this investigation due to the large variation in how and why it is applied [27,28].

Despite the notable advances in enrichment provisions in zoos, there are still significant limitations. While zoos may subscribe to the importance of providing goal-oriented enrichment, resources (staff time, experience, and funds for construction) and practical limitations mean that regular provision is not feasible [18,19,24,29]. Enrichment success over time is also often limited by anthropocentric designs: there are several examples of innovative and complex enrichment items where the lack of inherent variation and appropriate challenge leads to little or no animal engagement [18,30,31]. Another limitation of current enrichment practices aligns with the evolution in the understanding of how to enhance welfare itself, progressing from simply reducing negative affective states, to recognising the importance of concurrently promoting positive states. Enrichment strategies have also heavily focussed on eliminating sources of frustration (e.g., lack of space, incompatible social groups, lack of appropriate stimuli), while considerably less attention has been given to approaches which provide opportunities for problem-solving and cognitive challenges [5]. In order to allow animals choice and agency over their environment, which is a recognised goal for any modern zoo environment [18,32,33], environmental challenges must be “appropriate” in that they “may elicit frustration, but are potentially solvable or escapable through the application of cognitive and behavioural skills” [5]. Provision of appropriate challenges is sometimes described as “cognitive enrichment”, and this category of enrichment seems to be the least frequently applied in modern zoos [24], and only 3.5% of zoo enrichment studies (*n* = 744) analysed between 1985 and 2004 applied enrichment designed primarily to provide cognitive challenges [34].

Lastly, limitations in current enrichment approaches are clearly demonstrated when considering how foraging enrichment tends to be provided. For many animals, natural foraging behaviour is a complex sequence of activities that depend on cognitive mechanisms for their execution [5,9,14]. However, in practice, the main goal of foraging enrichment strategies tends to be simply increasing locomotion or time taken to consume food items, for example through scatter feeding or puzzle feeders. These goals do not address the protracted cognitive challenges associated with searching for, acquiring and manipulating cryptic, patchy or temporally variable food items [5]. A handful of published studies seem to have managed to incorporate these cognitive elements into the design of foraging enrichment devices or food-related cognitive tasks, [35,36,37,38], but although the common hypotheses, none were able to confirm that such enrichment led to multi-dimensional welfare improvements for the animals. Therefore, despite strong theoretical support, there is a lack of research on how cognitive enrichment impacts multi-dimensional, animal-based welfare measures.

Bottlenose dolphins are the most common cetacean species to be kept in zoos, and as gregarious generalist foragers with demonstrated problem-solving and cooperative hunting abilities, they are perhaps one of the species that would most benefit from cognitive enrichment. Globally, the facilities that generally house them are either open-system natural lagoons or artificial closed-system pools with few structures, and current or substrate variations. Both types of facilities predominantly manage the animals in the same style, where they are fed their daily food ration of (dead) fish in a series of positive reinforcement training sessions. These involve the trainer asking the animal to perform an already learnt or sometimes novel task and reinforcing the desired behaviour with a fish [39].

Similar to terrestrial zoo animals, enrichment practices for dolphins have improved in their effectiveness in increasing the quality or value of the environment to the animals [39,40,41,42,43,44,45]. However, the inventories of enrichment devices and strategies reported as part of facility programs indicate that the vast majority of items are simple and floating [39,41,46]. Cognitive enrichment for dolphins is not commonly applied [30]: a large multi-facility study surveyed 40 dolphin “habitats” in 38 facilities and found that indeed simplistic floating items were most often used, and less than 10% of habitats reported ever having used cognitive devices (“puzzle feeders”) [47]. The same study found that only around 25% of habitats provided food-based enrichment devices to dolphins (“feeder balls” and “puzzle feeders”) [47]. Three examples of cognitive foraging devices for dolphins exist in the literature but are reported as pilot studies and call for further investigation into the impacts on welfare [44,48,49]. One of these cognitive devices was recently paired with longitudinal social behaviour data in a study that found affiliative behaviour increased and agonistic behaviour decreased on days when enrichment was provided [25].

Improvement of dolphin welfare in zoos is in the public zeitgeist as evidenced by recent legislative changes and enduring public concern [20,50]. In the last decade, research has expanded significantly regarding the welfare assessment of cetacean species, with the most successful studies focussing on validating sets of multi-dimensional measures [47,51,52,53,54,55,56]. In many other zoo species, providing more foraging or food-based enrichment has been shown to improve welfare, where it is often reported to successfully decrease levels of abnormal repetitive behaviours (ARBs) among other parameters. Such studies have found that foraging enrichment leads to either a reduction in oral ARBs (those involving repetitive mouth movements, or regurgitation/ingestion) [10,12,13,17,26,57,58], or a reduction in locomotory ARBs which likely arising from thwarted “appetitive search behaviours” [8,9,14], or a decrease in both types of ARBs [9]. The aforementioned multi-facility study found that higher enrichment diversity was correlated with lower levels of dolphin pattern swimming [59]. ARBs have been very little studied in zoo-housed dolphins, but a variety of oral and locomotory ARBs are documented to exist [30,41,51,54,60,61]. There are no studies to our knowledge on providing dolphins with cognitive foraging enrichment versus other types of enrichment and documenting the impact on welfare.

The current study used a within-subject design on 11 bottlenose dolphins at Kolmårdens Djurpark, Sweden, and measured how several welfare indicators differed between two treatments of “cognitive” foraging enrichment and “non-cognitive” simple enrichment (also paired with food). Animal-based welfare indicators were selected based on their partial validation in other studies and included parameters pertaining to the enrichment provision sessions themselves as well as the rest of the animals’ time and daily activities outside the enrichment sessions. Our hypothesis was that the provision of cognitive foraging enrichment would improve several welfare parameters that would persist over the rest of the week.

## 2. Materials and Methods

### 2.1. Study Individuals and Site

Our study involved 11 bottlenose dolphins (*Tursiops truncatus*) housed at Kolmårdens Djurpark, Sweden (see Table 1 for age and sex characteristics). During the study, the dolphins inhabited two indoor interconnected pools: the “show pool” with a total surface area of 800 m^2^, and the “medical pool” (which had a lifting bottom) with a surface area of 180 m^2^. During the majority of the study, the dolphins had access to both pools: for 95% of the study (daytime hours) they had access to both pools, and for 5% of the time they had access only to the show pool. Dolphins were always kept as a group of 11. A third lagoon with a surface area of 900 m^2^ was behind a wall (connected by an underwater channel to the medical pool) and closed to the dolphins during the study as it was undergoing some renovations. The renovations had already been going on for 12 months before our study started. In order to assess whether the renovation work had any impact on our results, we measured several relevant parameters: the presence of noise from general construction work, and whether there was more intense drilling noise occurring. For 55% of the study period (daytime hours), dolphins were exposed to noise from construction work and for 18% of time they were exposed to noise from drilling.

All dolphins were fed a variety of fish during six to ten daily feeding sessions, which involved asking the animals tasks and rewarding them through positive reinforcement training. A total of 12 different caretakers looked after the dolphins, with about 5 scheduled at the park each day. Outside of the study period, the caretakers at Kolmårdens Djurpark provide the dolphins with a range of enrichment items multiple times per day. The average water temperature during the study was 22.4 °C.

### 2.2. Enrichment Treatments

The study was carried out for eight weeks, from November 2020 to January 2021. Two “treatments” were applied to the dolphins on an alternating weekly schedule during this period. In one treatment, the dolphins received non-cognitive enrichment, consisting of simple items (Figure 1) designed to encourage animals to touch, push or rub the device (hereafter called “non-cognitive enrichment”. The second treatment involved the dolphins being provided with enrichment devices that were more complex (Figure 1), designed to encourage problem-solving foraging behaviours focussed on finding and acquiring fish hidden within them (hereafter called “cognitive enrichment”). Each day, one of the fourteen enrichment devices in either treatment group (usually with several copies of the device to reduce competition) was given to the dolphins in the morning for approximately 1 h, and a second item was given in the afternoon for approximately 2 h. A fixed quantity of fish (0.1–0.2 kg per animal per day) was always hidden within the cognitive enrichment items, and the caretakers provided the same amount of fish when giving the non-cognitive enrichment (by laying the fish on top of the item if possible or throwing it next to the items), in order to control for the reinforcing value of acquiring a food item. An amount of 0.05 kg of gelatine per animal was provided every day with cognitive and non-cognitive enrichment items. On average, 1.6 kg of fish and gelatine was provided per animal per week during the cognitive treatment, and 1.64 kg of fish per animal during the non-cognitive weeks. Enrichment devices were placed in random locations in the pools (apart from those that had to be attached to specific areas) and not repeated again for the following 3 days in either treatment. All dolphins were familiarised with the enrichment devices for at least three months before the study. Dolphins were recognised individually by their trainers and the observer using visual indicators such as specific marks and body features.

### 2.3. Qualitative Data Collection

Qualitative data were collected during enrichment and training events by the 12 dolphin caretakers. In previous studies, qualitative data generated by dolphin and other zoo animal caretakers have been shown to be valid, accurate and reliable in terms of capturing holistic features such as behaviour and welfare [52,62,63,64,65].

#### 2.3.1. Enrichment Engagement Score

Positive enrichment engagement has been shown to correlate to play and other markers of improved dolphin welfare [39,41,46,66,67]. The dolphin caretakers observed the animals for at least five minutes and up to 15 min in order to evaluate individual dolphins’ engagement with each enrichment item provided. They qualitatively assigned an enrichment engagement score using a 5-point Likert scale, with the integers representing incremental grades of the dolphins’ interest towards and involvement with the enrichment (Figure 2). A score training protocol was applied before the study started where each score’s levels were described in detail and several practice sessions were given to promote concordance between caretakers.

#### 2.3.2. Willingness to Participate (WtP) in Training Sessions

Each dolphin’s willingness to participate (WtP) in every training session during the 8-week study was assessed by the trainers using a 5-point Likert scale. WtP was measured because it has been shown to correlate to other dolphin welfare parameters in several studies and therefore may be a valid indicator [64,68,69]. Training sessions were any events where the trainers asked known and novel tasks of the dolphins and fed them fish to reinforce certain behaviours. For this study’s purposes, training sessions never crossed over or included enrichment sessions and vice versa. During each training session, a different type of task tended to be focused on and included the following: husbandry behaviours, show-related behaviours, gating (practising moving pools), maintenance (known, simple behaviours) and research behaviours. WtP score used in this study was a focal animal 5-point Likert scale, with the integers representing incremental grades of the dolphin’s motivation during training sessions (Figure 3).

### 2.4. Quantitative Data Collection

#### Behavioural Observations

Behavioural observations were conducted by one trained observer between 06:00 and 19:00. A 5 min focal observation protocol was established with scan sampling every 15 s where the behaviour being performed was noted. We developed a behavioural repertoire containing ten behaviours (synchronous swimming, social play, sexual behaviour, agonistic behaviour, anticipatory behaviour, pattern swimming, stereotypic behaviour, other behaviours, alone enrichment interaction and social enrichment interaction) that had been shown the most evidence in the literature as indicators of welfare for this species (Table 2). “Other behaviours” were included in the ethogram to ensure that we were capturing all behaviours but were not statistically analysed since they were likely not linked to welfare.

### 2.5. Statistical Analyses

Statistical analyses were completed with the software R, version 4.2.1 [82]. To investigate whether the dolphin engagement with the enrichment differed between cognitive and non-cognitive treatment weeks, and to test whether dolphin WtP scores for training sessions differed between treatments, we applied cumulative link mixed-effects models using the “ordinal” package in R [83].

In terms of the quantified data taken in behaviour observations, we conducted multifactorial modelling to analyse whether there were differences between treatments, where we ran separate models for each dependent variable. Depending on the distribution of the data for each welfare-related behaviour, we used different models for the response variables. For the response variables “synchronous swimming” and “social play”, we applied a generalized linear mixed-effects model (GLMM) for proportional data (with a logit link) using “lme4” package in R [84]. For the response variable “stereotypic behaviour” we applied a zero-inflated generalized linear mixed-effects model (ZIGLMM) for count data using “glmmTMB” package in R [85]. For the response variables “sexual behaviour”, “agonistic behaviour”, “anticipatory behaviour”, “pattern swimming”, “alone enrichment interaction”, and “social enrichment interaction”, we applied a GLMM for count data with negative binomial distribution using “glmmTMB” package in R [85]. The data for all response variables were expressed as percentages (for analysis: frequency) of scans per total visible scans of the different behaviours observed in each 5-min focal observation.

For all models, we included the identity of the dolphin as a random factor to account for repeated measurements from the different animals. Predictors included the type of treatment (cognitive or non-cognitive), week number, the dolphin’s age, the presence of noise from construction work (yes or no), and the presence of noise from drilling (yes or no). For the WtP models, we included the type of session (gating, husbandry, maintenance, research or show) and pool access (show pool or both pools) as fixed factors and trainer identity as a random factor. For the enrichment engagement model, we also included the duration of caretaker observation (how long they observed the enrichment session), time of day (morning or afternoon) and pool access as fixed factors. For the quantitative behavioural models, we included the presence of enrichment (yes or no) and pool access as fixed factors.

Assumptions were verified by plotting residuals versus fitted values with the package “DHARMa” [86]. For GLMMs models showing signs of overdispersion, we included a case-level random factor. Furthermore, collinearity was tested for all models via the variance inflation factor (VIF) [87]. Statistical significance of the fixed effects and their interaction with the response variables was determined in all cases using Wald chi-square tests. For each model, post hoc analysis was completed by conducting multiple pairwise comparisons of the estimated marginal means between cognitive and non-cognitive treatments using “EMMEANS” package in R [88]. As multiple comparisons were performed to test for significant differences between treatments in each behavioural observation, it was necessary to control for type I errors. Hence, as an alternative to Bonferroni correction [89,90], we relied on the procedure introduced by Benjamini and Hochberg [91], which is similar to Bonferroni’s, but also reduces type II errors by controlling for the false discovery rate [92].

To evaluate whether there was a significant habituation effect with the enrichment devices during the eight weeks of the study, we performed a Regression Fitted Line Plot between the percentage of scans for “alone enrichment interaction” and “social enrichment interaction” and the treatment week numbers.

## 3. Results

### 3.1. Enrichment Engagement Scores

During the study period, 1121 qualitative enrichment engagement scores were assigned by the caretakers, an average of 102 per dolphin. A total of 538 scores were assigned to interactions during the cognitive enrichment treatment, and 583 during the non-cognitive treatment. According to these scores, the dolphins were significantly more engaged with the enrichment devices during the cognitive treatment, indicated by more “highly engaged” scores and fewer “no interest” scores, compared to during the non-cognitive treatment (Figure 4, Table 3). Enrichment engagement scores were affected by the time of day, the duration of observations, what pools the animals had access to, and the presence of construction work (Table 4).

### 3.2. Dolphin Willingness to Participate in Training Sessions

Over the study, the dolphins’ caretakers assigned 5843 willingness to participate (WtP) in training sessions scores, an average of 498 per dolphin. A total of 2813 scores were given during the cognitive enrichment treatment, and 2670 during the non-cognitive treatment. Dolphins’ WtP in sessions with their trainers differed significantly between enrichment treatments, with dolphins showing higher motivation to participate in training sessions during the cognitive enrichment provision (Figure 5, Table 3). Dolphins’ WtP scores were also significantly higher when they had access to both pools (Table 4).

### 3.3. Behavioural Differences between Treatments

A total of 704 5 min focal behavioural observations were conducted during the study, an average of 64 per dolphin or 320 min. During both cognitive and non-cognitive enrichment treatment weeks, 352 observations were taken on the group in total. The dolphins interacted significantly more with enrichment devices during the cognitive versus the non-cognitive treatment (Figure 6, Table 3).

No significant differences were found between cognitive and non-cognitive treatments in the behaviours “synchronous swimming”, “social play”, “agonistic behaviour”, “sexual behaviour” and “pattern swimming” (Figure 7, Table 3). However, we found a significant difference between treatments in the expression of “anticipatory” and “stereotypic” behaviours. Dolphins performed less anticipatory and stereotypic behaviours during the cognitive enrichment treatment than during the non-cognitive treatment (Figure 7, Table 3).

### 3.4. Effects of Different Environmental Factors on Behaviour

We also found several significant effects of some of the environmental factors on behavioural frequencies (Table 5). The week number and age significantly affected the frequency (in terms of percentage of scans out of total visible scans per observation, hereafter described as frequency) of alone enrichment interaction ((a) in Table 5), where more frequently alone interactions with items were seen by younger dolphins, and with increasing week number as the study progressed. Social enrichment interaction was significantly increased with week number as well ((b) in Table 5). Dolphins showed significantly higher frequencies of synchronous swimming when enrichment was not present in the pool ((c) in Table 5). Age, drilling noise, pool access, enrichment presence and drilling noise affected the duration of social play ((d) in Table 5). Dolphins showed significantly less social play the older they were, during periods of drilling noise, when they had access to both pools, and when enrichment was present. Sexual behaviour was significantly lower when enrichment was present and during periods of noise caused by construction work and drilling ((e) in Table 5). Dolphins performed significantly less agonistic behaviour when enrichment was present in the pool ((f) in Table 5). The frequency of anticipatory behaviour was affected by week number, dolphins’ age, enrichment presence and noise from drilling ((g) in Table 5). Dolphins performed this behaviour significantly more the younger they were, in the earlier weeks of the study, during periods of noise caused by drilling and when enrichment was present. Lastly, dolphins showed significantly higher frequencies of pattern swimming but lower levels of stereotypic behaviour during the presence of drilling noise, and higher frequencies of stereotypic behaviour during periods of general construction noise ((h) and (i) in Table 5).

### 3.5. Habituation

We did not find evidence of habituation effects. Conversely, we found a significant increase in the frequency of alone enrichment interaction (X^2^ = 11.06; df = 1; *p* = 0.0009) and social enrichment interaction (X^2^ = 23.86; df = 1; *p* = ≤ 0.001) during the eight weeks of the study.

## 4. Discussion

During the weeks that cognitive as opposed to non-cognitive enrichment was provided to the bottlenose dolphins, we found that the animals engaged more with the cognitive enrichment as measured through multiple parameters. The dolphins were also more motivated to participate in training sessions and performed less anticipatory and stereotypic behaviours. These results imply that cognitive enrichment provision improved several indicators of dolphin welfare in both an immediate and longer-term timeframe and allowed us to consider implications for other dolphins’ welfare in zoos.

### 4.1. Treatment Effects on Behaviour during Enrichment Provision

The two treatments applied in this study were the provision of cognitive foraging enrichment items twice a day for a week, followed by a week of non-cognitive enrichment items, and these treatments were alternated for 8 weeks continuously for the same group of 11 dolphins. We observed both the first- (direct) and second-order effects of these treatments on the animals’ welfare. Direct or shorter-term effects of the treatment were measured through the dolphins’ engagement with the enrichment itself and using three parameters: qualitative caretaker scores of enrichment engagement, and behavioural observations which documented the frequency of both alone and social enrichment engagement. We found significant differences in all three parameters where engagement was higher during cognitive enrichment weeks (Figure 4 and Figure 6). This result demonstrates that although qualitative, the caretakers’ scores were accurate as they correlated with quantified behavioural data, and, in agreement with previous literature [62,63,64], suggests these more practical types of scores are appropriate for caretakers to use in monitoring the welfare of zoo animals such as dolphins.

The same amount of fish was provided with every enrichment item in both cognitive and non-cognitive enrichment scenarios, allowing us to conclude that the increased engagement with the cognitive enrichment was not due to the provision of food in the foraging device. The literature suggests that the difference shown in our results may be linked to the contrafreeloading principle, which describes the phenomenon of animals seemingly preferring to work for food even when the same items are freely available. Given the fact that contrafreeloading contradicts optimal foraging theory and has not been ubiquitously observed across species and contexts, many argue its occurrence is due to secondary reinforcers associated with the reward acquisition, such as information gathering, positive affective states or performing species-typical behaviours [93,94,95]. A previous study on dolphins’ response to a cognitive foraging enrichment item found evidence of contrafreeloading and suggested the item had intrinsic rewarding properties [48].

Foraging (locating and consuming prey) takes up about 60–70% of wild bottlenose dolphins’ activity budget [96,97,98,99]. In zoological facilities, dolphins tend to receive nearly 100% of their food from training sessions, usually taking up about 2 h in total of the 24 h day ([47], I. Clegg, personal observation), which is only 8% of their activity budget. While training sessions can sometimes be cognitively challenging, not all are, and the dolphins have restricted agency about how they acquire the food reward [20,27,28]. Enrichment devices designed to stimulate dolphins’ cognitive abilities related to foraging may provide them with agency, and explorative and problem-solving opportunities that they do not experience in other areas of their lives [24,35,36], and may encourage species-typical behaviours. We conclude that the higher levels of dolphin engagement in our study were likely a result of some of these opportunities being provided by cognitive enrichment, whereas far fewer opportunities (although still some, as we saw some engagement) were provided by non-cognitive enrichment.

Given the voluntary nature of the behaviour and opportunities that the cognitive foraging items aimed to provide, it is likely that engagement with this enrichment generated some types of positive affective states in the dolphins. Our study cannot show how long these affective states may have lasted, or how meaningful they were, but the second-order welfare effects described below provide supporting evidence that the cognitive enrichment provision was correlated with positive welfare-related changes that continued over the whole treatment week.

### 4.2. Longer-Term Treatment Effects on Welfare-Related Behaviours during the Rest of the Week

We also found significant effects of the enrichment treatments on other welfare-related behaviours during the rest of the treatment week. Firstly, we found that the dolphins’ willingness to participate (WtP) in training sessions (all types) was higher during cognitive enrichment weeks than non-cognitive (Figure 5). WtP has been previously suggested as a welfare indicator for dolphins [64,68,69], but the correlations found in this study with cognitive enrichment engagement and other welfare parameters further validate it. WtP, as assessed qualitatively by caretakers, aims to measure the animals’ motivation to conduct positive reinforcement training exercises: although likely not a linear relationship, in general animals in better welfare are more likely to engage in opportunities with potential rewards [100,101,102]. While the link between enrichment engagement and training motivation has not been shown before in dolphins, a recent study on two captive harbour seals (*Phoca vitulina*) found that enrichment provision increased motivation for training sessions [103].

Secondly, we looked at the effects of the enrichment treatments on other welfare-related behaviours in the animals’ activity budget during the rest of the week. Cognitive enrichment provision significantly reduced the frequency of both anticipatory behaviour for training sessions, and stereotypic behaviour. While some anticipatory behaviour for predictable rewarding events reflects positive welfare, in excess it likely indicates negative states [101,104], and one study has demonstrated this with zoo-housed dolphins by linking anticipatory behaviour to cognitive bias [78]. If we pair the lower anticipatory behaviour finding with the higher WtP in training sessions in cognitive enrichment weeks, according to the reward-sensitivity theory the treatment did not diminish motivation for rewards but increased environmental stimulation [101]. Stereotypic behaviours are one of the most well-established indicators of reduced welfare, although their use in practical assessments can be complicated by the fact that they can become divorced from their original context and thus are not always linked to the current environment [105,106]. Stereotypies are very seldom reported in dolphin welfare research [30,107]: only one previous study has documented stereotypy reduction in dolphins previously, through the use of simple non-foraging enrichment [41]. Therefore, although in this study the overall stereotypic behaviour frequencies were very small (0.01% of the activity budget), the significant difference supports the finding that cognitive enrichment provision improved welfare. Second-order positive effects of cognitive enrichment on dolphin welfare versus other types of enrichment have not been reported before: two previous studies only considered welfare-related behaviours when the enrichment was present [44,49], and another did find potential welfare improvements in the presence or absence of certain behaviours even when cognitive enrichment was not present, but did not test the difference between cognitive and non-cognitive enrichment [25].

This study’s findings of lower frequencies of anticipatory and stereotypic behaviour, as well as the higher motivation for training sessions (WtP), after the cognitive foraging enrichment was removed indicate that its provision may have led to longer-term positive welfare states that continued over the treatment weeks. The non-cognitive enrichment was able to stimulate interest and engagement from the dolphins (Figure 4), but according to our results was not able to generate these sustained affective state changes, and therefore might be said to have fewer and/or weaker welfare benefits.

### 4.3. Limitations of the Study and Further Research

There are several limitations of our study worth taking into account. Only one group of dolphins (*n* = 11 animals) was involved, which means the results may only be applicable to them. Furthermore, we did not analyse individual differences in the welfare-related parameters, for which we may have found variance in response to the cognitive versus non-cognitive enrichment. A study on foraging enrichment in giraffes (*Giraffa camelopardalis*) found individual differences in contrafreeloading tendencies [108], and several studies have reported individual differences in bottlenose dolphin response to enrichment in general [39,109]. The welfare conclusions of this study are limited by our choice to use only behavioural welfare parameters, despite including several holistic, multidimensional behavioural indicators. Future research might prioritise collecting physiological and cognitive data to cover multiple dimensions of animal emotional states [110,111]. The qualitative data we collected may have been subject to bias since the dolphin caretakers’ made their ratings with the knowledge of which treatment week it was. It would also be interesting to continue investigating the impact of foraging enrichment on oral versus locomotory ARBs, which we only touched upon in this study by splitting ARBs into pattern swimming and other stereotypic behaviour.

Another limitation may have been the presence of construction work at the Kolmårdens Djurpark dolphin facility, which had been going on for a year before and continued during the study. We attempted to control for these factors as much as possible in our analysis and found several effects of the presence of drilling noise (more intense) in particular. The dolphins showed significantly less social play, more anticipatory behaviour, more stereotypic behaviour and more pattern swimming when drilling noise occurred (Table 5, indicating that there may have been some disturbance from the construction, as has been found in other studies on dolphins [72,112]. However, due to another study limitation of only 8 weeks of data collection, the correlations between the environmental variables and behavioural frequencies may not be reliable, particularly for behaviours that occur at low frequencies overall. For example, we found the contradictory result of lower levels of stereotypic behaviour during periods of noise caused by drilling, and higher frequencies of stereotypic behaviour during periods of general construction noise, likely caused by the fact that stereotypic behaviour occurred on average for only 0.01% of observed time. Nevertheless, although we found enrichment treatment effects while controlling for construction and drilling noise in our models, we may have found different results were we to have conducted this study when there was no construction work occurring.

Lastly, a limitation of our study, or perhaps more optimistically an area for future research, might be the question of how stimulating our “cognitive” foraging enrichment items actually were. Longer-term research would be invaluable in starting to answer this question. As Clark pointed out, it is difficult to define cognitive enrichment [24] and presenting animals with problems to solve is a delicate balance where to find an activity rewarding, they should perceive a challenge but also possess the skills to ultimately acquire a reward or rewarding experience [5,24,48]. The cognitive enrichment items in this study were designed through a goal-based approach [22], and although we could not confirm that they were perceived as cognitive challenges by the animals before we applied them, our first and second-order results strongly suggest that they were more stimulating than the non-cognitive enrichment that was applied. The items were not novel to the animals, and future studies might explore how novelty interacts with the cognitive element of enrichment. Further research might also measure parameters such as flow, agency and competence to better understand whether the items represent appropriate challenges for the dolphins [24].

### 4.4. Implications for Dolphin Enrichment and Welfare in Zoos

Our results could be used to make conservative suggestions about dolphin welfare in other zoos and looking forward could provide recommendations on how to improve welfare through cognitive foraging enrichment. These discussions could be very generally and cautiously applied to other zoo-housed species with similar cognitive abilities and foraging ecologies to dolphins, but would certainly warrant species-specific research.

Although we did not conduct a full welfare assessment on the dolphins in this study, we measured several welfare parameters in relation to the type of enrichment provided, finding that cognitive foraging enrichment led to immediate and longer-term welfare improvements, whereas non-cognitive enrichment did not. In general, dolphin facilities do not commonly provide cognitive and/or foraging enrichment items, instead focussing on simple, floating items without problem-solving or foraging components [20,30,41,46,47]. Although less complex items can be stimulating as well [113], and indeed in the current study the dolphins were engaged by the non-cognitive items to some extent, a variable inventory of items that promote a range of species-typical behaviours may be crucial to induce longer-term positive welfare states. Although seldom reported, oral abnormal repetitive behaviours (ARBs) have been documented for dolphins in zoos and include repetitive regurgitation, water throwing, bubble snapping, or foreign object ingestion [30,41,51]. ARBs, such as the stereotypic behaviours measured in this study, most often develop in response to boredom or frustration due to the lack of control and opportunities in their environment. We cannot conclude that dolphins at a facility that does not provide cognitive enrichment experience poor welfare, but since most cetacean exhibits are uniform and invariable [20,24,50], it may be safe to conclude that appropriate cognitive foraging enrichment items will very likely promote greater environmental stimulation, reduce ARBs and increase positive affective states for some animals in the group [17,24,105].

There are several practical and resource-related factors that likely constitute the main barriers to dolphin facilities providing cognitive foraging enrichment on a regular basis. Complex items are often harder to risk-assess from a safety standpoint and may require more time and staff for implementation. More staff time, skill, creativity and physical resources are usually needed to design and build cognitive enrichment items. For foraging devices where fish may be left in the environment, facility veterinarians often have a time limit of 15 min before the fish must be removed in case of bacterial proliferation (I. Clegg, personal communication), which can make it difficult to design appropriate foraging challenges. While valid, these challenges can all be overcome with careful design, communication and application of cognitive foraging enrichment. Although providing zoo animals with challenges poses risks and may result in reduced ability to control the animals, the alternative, sterile environment has no agency or opportunities to work for rewards and can result in equally or more significant welfare reductions [4,114]. Other management-based barriers to providing cognitive foraging enrichment may be a reluctance to use fish for enrichment as it is required for training sessions, including guest interactions or presentations. In addition, cognitive enrichment may be provided but some or all dolphins do not choose and/or understand how to engage with the item, which discourages caretakers from further provision [42,48]. In most cases, this can be overcome with a well-planned initial training schedule to reduce neophobic responses [42,46].

Based on the cognitive foraging enrichment program applied during this study, we recommend that cognitive foraging enrichment be goal-oriented [22], i.e., designed to target a specific behaviour or type of activity. The inventory of items should be large enough that items should be repeated only every three days or more: in the current study, we did not observe habituation to items over the study, and in fact, engagement increased with week number. The enrichment inventory can be made variable by including non-cognitive items, which could target other sensory, social, or exploratory behaviours. Focal animal engagement should be documented using either qualitative scales or quantitative methods so that individual preferences and the impact of environmental factors can be understood.

### 4.5. Next Steps for Dolphin Cognitive Foraging Enrichment

As with other species, technology is likely to play an important role in increasing the number of facilities providing dolphins with cognitive enrichment and advancing its effectiveness and therefore welfare benefits [18,31]. The well-designed and innovative use of technology allow caretakers and other staff to dissociate themselves from the enrichment items, which will lead to more variability and unpredictability, and consequently to control and agency by the animals and then to self-regulation, flow and competence [18,24]. Technology, such as animal–computer interaction (ACI), has been successfully used in laboratory settings to test the cognitive abilities of primates and is starting to be applied in zoo contexts as well [18,38,115]. While the lack of digits is an obvious initial barrier to traditional ACIs such as touchscreens, an innovative device has already been piloted with dolphins at the current study’s Kolmårdens Djurpark: an interactive screen where the dolphins’ echolocation beams are transmitted into a “paint” function, based on the intensity of the acoustic signal [29]. Symbol keyboards have also been used successfully in the past to enable dolphins more choice over which enrichment items, activities and environments they would like to experience [45,116]. A mechanical feeder ball that is able to be manipulated and whose food release can be programmed to promote different foraging behaviours have been trialled with an eastern black rhinoceros (*Diceros bicornis michaeli*) [36], and automated feeders can encourage more natural feeding patterns and time budgets for grazers and browsers in particular [18].

Lastly, the animal-based goals for cognitive enrichment should evolve to encourage more “sophisticated” forms of problem-solving: trial-and-error learning is the least refined and easiest to encourage with enrichment, progressing to mental representation and future planning [48]. For the highly social dolphin species, planning skills in the wild are paired with projections of conspecific behaviour and social learning, and this phenomenon should particularly be targeted with future cognitive foraging enrichment efforts. At Kolmårdens Djurpark, the current study’s facility, social learning was anecdotally reported before the study started: when the fake kelp plants attached to the bottom were first presented, the dolphins tried several ways to acquire the fish hidden inside them. One animal discovered an effective technique was to beat her tail right on top of the plant before spinning around to grab the fish, and soon after, all animals in the group also used this method. While this example probably represents a less complex trial-and-error approach followed by social learning from the group, it mirrors the cultural sharing of foraging techniques documented in wild cetaceans [117,118,119] and is therefore a meaningful outcome of cognitive foraging enrichment provided in zoo settings.

## 5. Conclusions

For the first time, dolphin welfare impacts of cognitive versus non-cognitive enrichment were investigated, both in the immediate and longer-term after enrichment was removed. As a group of 11 dolphins, it was found that they interacted significantly more with the cognitive than non-cognitive devices, both when measured through qualitative caretaker scores and direct behavioural observation. During the rest of the treatment week, during cognitive enrichment weeks, the dolphins were also significantly more motivated to participate in training sessions and showed less anticipatory and stereotypic behaviour. These changes in welfare-related behaviours suggest that the provision of cognitive enrichment improved longer-term affective states and potentially overall welfare, whereas non-cognitive enrichment did not. The results of this study and our recommendations could be used to inform enrichment management for dolphins in zoos, where currently cognitive foraging enrichment does not seem to be provided often. Our findings may be applicable to other similar species that show advanced problem-solving abilities, and we discuss how zoo cognitive enrichment practices might evolve to more effectively promote positive welfare.

## Figures and Tables

**Figure 1 animals-13-00238-f001:**
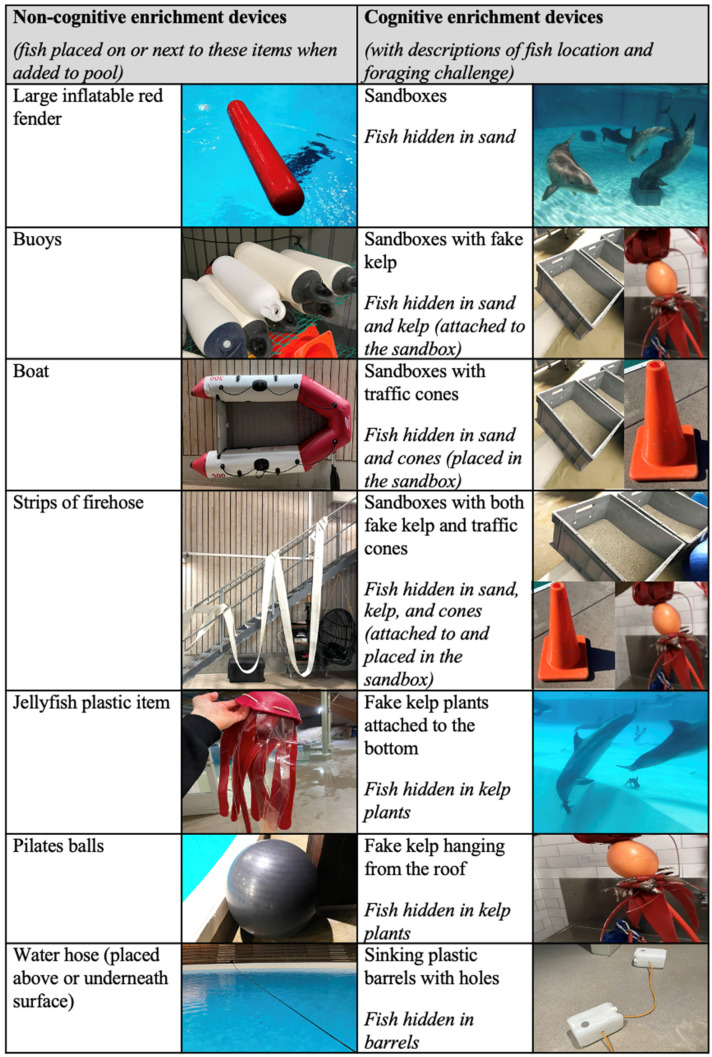
List of 14 devices provided during each of the non-cognitive and cognitive enrichment treatment weeks during the study.

**Figure 2 animals-13-00238-f002:**
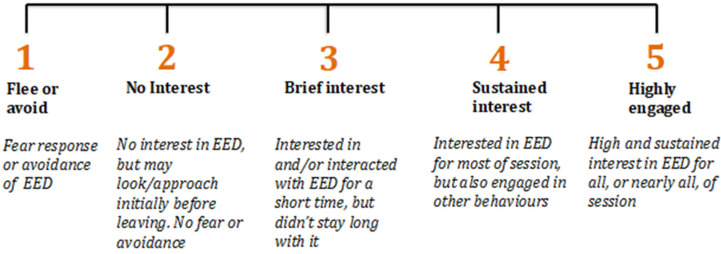
Five-point Likert scale used by caretakers to score each dolphin’s engagement with enrichment items every time enrichment was provided over the 8-week study.

**Figure 3 animals-13-00238-f003:**
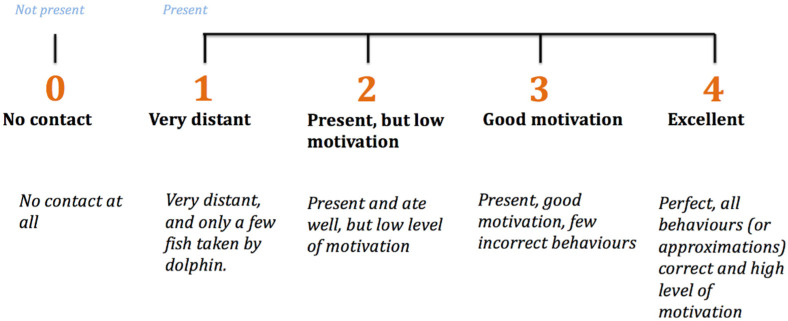
Five-point Likert scale (from [64]) used by trainers to score each dolphin’s willingness to participate (*WtP*) in daily training sessions over the 8-week period of the study.

**Figure 4 animals-13-00238-f004:**
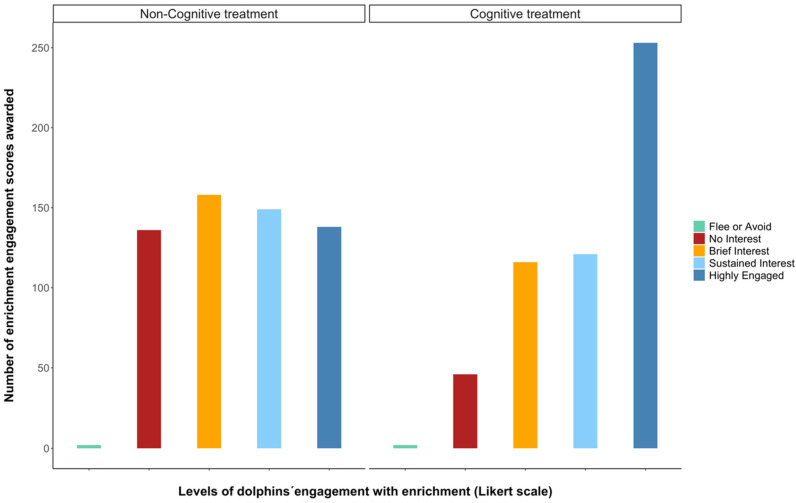
Dolphins´ enrichment engagement scores as scored by their caretakers. Coloured bars represent the number of times an enrichment engagement score (on a 1–5 Likert scale, Figure 2) was attributed to dolphins over the 8-week experiment.

**Figure 5 animals-13-00238-f005:**
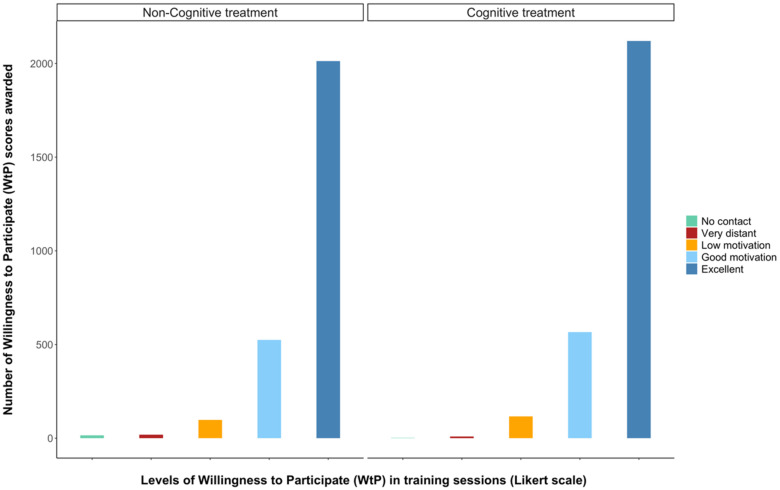
Comparison of caretaker scores for willingness to participate (WtP) in training sessions between cognitive and non-cognitive treatments. Coloured bars represent the number of times a WtP score (on a 0–4 scale) was attributed to the dolphins.

**Figure 6 animals-13-00238-f006:**
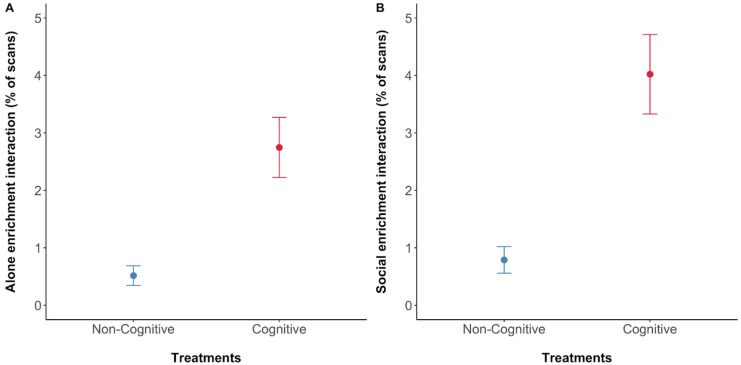
Dolphin’s enrichment interaction: (**A**) percentage of scans for individual interaction with the enrichment; (**B**) percentage of scans for group interaction with the enrichment. Percentage of scans denotes the average percentage of scans where the behaviour was performed as a proportion of the total visible scans in the focal 5 min observations. Means are represented by filled dots accompanied by standard error bars.

**Figure 7 animals-13-00238-f007:**
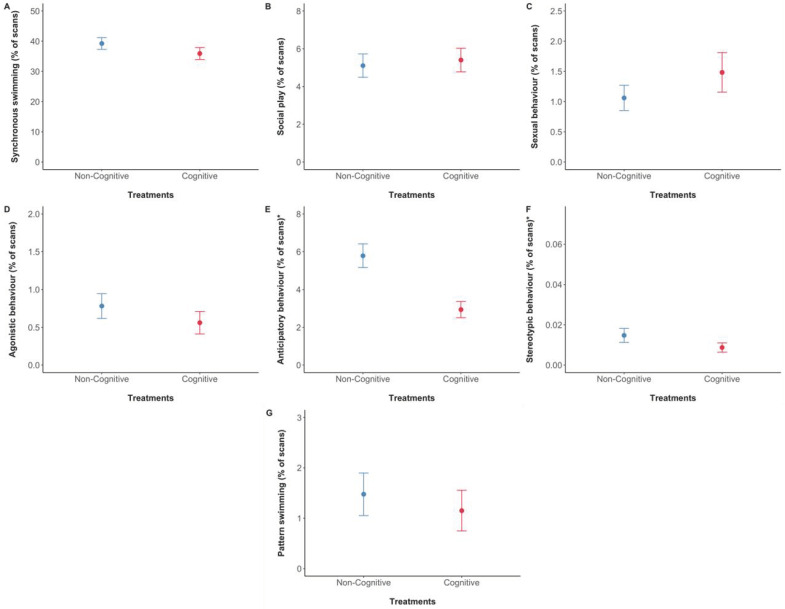
Percentage of behaviours displayed by the dolphins between cognitive and non-cognitive treatments. (**A**) Synchronous swimming; (**B**) social play; (**C**) sexual behaviour; (**D**) agonistic behaviour; (**E**) anticipatory behaviour; (**F**) stereotypic behaviour; (**G**) pattern swimming. Percentage of scans *y*-axis labels describe the percentage of scans where the behaviour was performed out of the total visible scans in the focal 5 min observation. Means are represented by filled dots accompanied by standard error bars. Asterisks (*) by plot names indicate which behavioural frequencies were significantly different between treatments.

**Table 1 animals-13-00238-t001:** Eleven Bottlenose dolphins (*Tursiops truncatus*) at Kolmårdens Djurpark included in the study.

Name	Sex	Age at Time of Study
Alana	Female	3
Neptun	Male	3
Finn	Male	3
Peach	Female	6
Pärla	Female	9
Fenah	Female	13
David	Male	17
Luna	Female	20
Lyra	Female	22
Ariel	Female	25
Nephele	Female	38

**Table 2 animals-13-00238-t002:** Welfare-related bottlenose dolphin behaviours (with definitions and references showing link to welfare) included in the study’s ethogram for the behavioural observation protocol.

Behaviour	Definition	References
Synchronous swimming	Dolphin swimming within one body length of another dolphin, showing parallel movements and body axes, with only a few seconds delay between movements	[53,70,71,72]
Social play	Dolphin engaged in rubbing, nudging, chasing, pushing, and jumping, all more gently and at lower speeds than agonistic interactions (and no enrichment items involved)	[73,74,75]
Sexual behaviour	Dolphin engages in genital-to-genital contact, touches a genital to another dolphin’s body part, or touches its body part to another dolphin’s genitals	[51,55,71,76]
Agonistic behaviour	Social behaviours that may include tracking another dolphin, S-postures, chasing, biting, body slamming, jaw clapping, which occur at higher speeds than play (around 4 m/s, skin usually observed to be rippling)	[51,55,71,73,77]
Anticipatory behaviour	Dolphin is looking above water towards the pool sides, usually at the location where training sessions start or where people are visible. Includes surface looking (either stationary or while swimming), spy hopping and underwater looking (clearly looking up at the poolside from underwater).	[52,78,79]
Pattern swimming	Dolphin swims in the same pattern, which does not vary from the route and may breathe in the same place. The dolphin is not alert or vigilant (e.g., looking around)	[54,59,60,80]
Stereotypic behaviour	Any repetitive behaviour that is invariant, repetitive and has no apparent function.	[41,51,61,81]
Alone enrichment interaction	Solitary interest (looking at item while 1 body length or less away) and play behaviours (e.g., pushing, carrying, biting, rubbing) involving an enrichment item	[41,46,66,67]
Social enrichment interaction	Group interest (looking at item while 1 body length or less away) play behaviours (e.g., pushing, carrying, biting, rubbing) involving an enrichment item	[41,46,66,67]
Other behaviours	Any other behaviour	

**Table 3 animals-13-00238-t003:** Pairwise comparisons of model-estimated marginal means (EMMs) between non-cognitive and cognitive treatments for: enrichment engagement scores, willingness to participate (WtP) scores, and behavioural observations. Significant effects and variables are highlighted in bold; asterisks indicate behavioural differences remaining significant after Benjamini and Hochberg’s (1995) correction.

Dependent Variable	Estimate	Standard Error	*p*-Value
**Enrichment engagement scores**	−1.17	0.14	**<0.0001**
**WtP scores**	0.281	0.0873	**0.0013**
**Alone enrichment interaction**	−1.74	0.436	**0.0001 ***
**Social enrichment interaction**	−0.9	0.283	**0.0015 ***
Synchronous swimming	0.48	0.256	0.0606
Social play	−0.284	0.253	0.2616
Sexual behaviour	−0.232	0.254	0.3611
Agonistic behaviour	0.431	0.37	0.2447
**Anticipatory behaviour**	0.414	0.149	**0.0056 ***
**Stereotypic behaviour**	0.628	0.189	**0.0010 ***
Pattern Swimming	0.298	0.404	0.4603

**Table 4 animals-13-00238-t004:** Effects of treatments (non-cognitive vs. cognitive), week (week number of experiment), pool access (only show pool vs. both pools), age, general construction noise (presence vs. absence), drilling noise (presence vs. absence), type of session, duration of caretaker observations and time of day on enrichment engagement scores and WtP scores. Analyses by cumulative link mixed-effects models for ordinal data. Dolphin identity (a, b) and trainer identity (a) were included as random factors. Significant differences (*p* < 0.05) and variables are highlighted in bold.

Predictors		Treatments	Drilling Noise	Construction Work Noise	Age	Pool Access	Week	Type of Session	Duration of Observations	Time of Day
**(a) Enrichment engagement scores**	*X* ^2^	75.09	2.94	3.93	1.58	20.7	1.56	-	5.19	7.99
	*p*	**<0.001**	0.08	**0.047**	0.20	**<0.001**	0.21	-	**0.02**	**0.004**
**(b) Willingness to Participate**	*X* ^2^	10.41	0.69	0.21	0.007	60.6	0.56	3.41	-	-
	*p*	**0.001**	0.40	**0.64**	0.93	**<0.001**	0.45	0.49	**-**	**-**

**Table 5 animals-13-00238-t005:** Effects of treatments (non-cognitive vs. cognitive), week (week number of experiment), pool access (only show pool vs. both pools), age, enrichment (presence vs. absence, general construction noise (presence vs. absence) and drilling noise (presence vs. absence) on the occurrence of different behaviours of bottlenose dolphins. Multifactorial analyses by GLMM for proportional data (with a logit link) (c, d), by ZIGLMM for count data (h) and by a GLMM for count data with negative binomial distribution (a, b, e, f, i), all of these including dolphin identity as a random factor. Significant differences (*p* < 0.05) are highlighted in bold.

Predictors		Treatments	Drilling Noise	General Construction Work Noise	Enrichment Presence	Age	Pool Access	Week Number
(a) Alone enrichment interaction	*X* ^2^	15.85	0.9	0.53	-	16.89	0.6	3.99
	*p*	**<0.001**	0.34	0.46	-	**<0.001**	0.43	**0.04**
(b) Social enrichment interaction	*X* ^2^	10.12	3.26	0.03	-	2.2	0.02	14.34
	*p*	**0.001**	0.07	0.85	-	0.13	0.87	**0.0001**
(c) Synchronous swimming	*X* ^2^	3.52	0.01	0.07	9.09	0.11	1.47	1.35
	*p*	0.06	0.92	0.78	**0.002**	0.73	0.22	0.24
(d) Social play	*X* ^2^	1.26	9.96	3.22	5.53	74.92	3.79	0.007
	*p*	0.26	**0.001**	0.07	**0.02**	**<0.001**	0.051	0.93
(e) Sexual behaviour	*X* ^2^	0.83	6.62	7.87	4.5	1.38	0.34	1.48
	*p*	0.36	**0.01**	**0.005**	**0.03**	0.23	0.55	0.22
(f) Agonistic behaviour	*X* ^2^	1.35	3.12	1.27	4.94	3.66	0.1	0.68
	*p*	0.24	0.08	0.25	**0.03**	0.06	0.74	0.4
(g) Anticipatory behaviour	*X* ^2^	7.73	4.27	0.02	4.99	4.12	0.46	23.55
	*p*	**0.005**	**0.04**	0.88	**0.03**	**0.04**	0.49	**<0.001**
(h) Stereotypic behaviour	*X* ^2^	11.01	5.04	4.51	0.26	0.52	0.35	3.65
	*p*	**0.0009**	**0.02**	**0.03**	0.6	0.46	0.55	0.06
(i) Pattern swimming	*X* ^2^	0.54	4.08	2.18	0.1	1.24	2.46	0.71
	*p*	0.46	**0.04**	0.14	0.74	0.26	0.11	0.39

## Data Availability

The data presented in this study are available on request from the corresponding author.

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
