# Peer review of "Cognitive Foraging Enrichment (but Not Non-Cognitive Enrichment) Improved Several Longer-Term Welfare Indicators in Bottlenose Dolphins"

_animals, 2023, doi:10.3390/ani13020238_

Round 1

Reviewer 1 Report

An interesting manuscript, written with the correct style of a scientific article, however, due to the duty of the reviewer, I would like to draw the authors' attention regarding:

Was the introduction of so many cognitive enrichments (of various types) in a short time necessary to obtain clear results? Wasn't the curiosity of animals resulting from the reaction to the NEW ONE the dominant result?  It is worth enriching the discussion with answers to these questions.

Graphs 3 and 4 are not precise enough to read due to the scale of the graph, you need to add numeric values next to each column.

For Figures 5 and 6, there is not enough legend described: is the range for extreme values (max-min)?

The table is about comparing elements in the same relationship, so the Pairwise comparison  column seems redundant, and the introduction of this relationship should be included in the title of the table.

Reviewer 2 Report

This is a well-done study that focuses on an important issue - the welfare of captive dolphins. The authors do the service of pointing out that - in most captive facilities - the dolphins are not engaged at a level that is either naturalistic or complex.  As someone who has studied their brains and intelligence for thirty years I appreciate any attempts to stimulate their minds in an environment that can never match a free-ranging situation.  I think the authors did a good job of balancing the factors at play and also relating the results. They were also honest in noting the inherent limitations of such a study. One thing I'd like to see is a longer-term study of these kinds of complex naturalistic enrichment devices. It may be that novelty is a bigger part of the results than one can determine with a short study time.  In the end, the ultimate question is whether any enrichment device can provide what dolphins need in order to thrive in captivity.  That is a question that the authors also pose. The methodology is very well thought out and articulated.  I don't have any major problems with this study.  Well done!

Reviewer 3 Report

I have no correction or changes to suggest. I find your study very interesting and helpful, I hope you can implement it in the future.
